# Regulation of Bone Cell Differentiation and Activation by Microbe-Associated Molecular Patterns

**DOI:** 10.3390/ijms22115805

**Published:** 2021-05-28

**Authors:** Yeongkag Kwon, Chaeyeon Park, Jueun Lee, Dong Hyun Park, Sungho Jeong, Cheol-Heui Yun, Ok-Jin Park, Seung Hyun Han

**Affiliations:** 1Department of Oral Microbiology and Immunology, and Dental Research Institute, School of Dentistry, Seoul National University, Seoul 08826, Korea; kwonykag@snu.ac.kr (Y.K.); chaeyeon@snu.ac.kr (C.P.); wndms0013@snu.ac.kr (J.L.); pdh5958@snu.ac.kr (D.H.P.); ssgd33@snu.ac.kr (S.J.); 2Department of Agricultural Biotechnology, and Research Institute of Agriculture and Life Sciences, Seoul National University, Seoul 08826, Korea; cyun@snu.ac.kr

**Keywords:** bone diseases, bone homeostasis, bacteria, microbe-associated molecular patterns, osteoblast, osteoclast, pattern-recognition receptors, secretory microbial molecules

## Abstract

Gut microbiota has emerged as an important regulator of bone homeostasis. In particular, the modulation of innate immunity and bone homeostasis is mediated through the interaction between microbe-associated molecular patterns (MAMPs) and the host pattern recognition receptors including Toll-like receptors and nucleotide-binding oligomerization domains. Pathogenic bacteria such as *Porphyromonas gingivalis* and *Staphylococcus aureus* tend to induce bone destruction and cause various inflammatory bone diseases including periodontal diseases, osteomyelitis, and septic arthritis. On the other hand, probiotic bacteria such as *Lactobacillus* and *Bifidobacterium* species can prevent bone loss. In addition, bacterial metabolites and various secretory molecules such as short chain fatty acids and cyclic nucleotides can also affect bone homeostasis. This review focuses on the regulation of osteoclast and osteoblast by MAMPs including cell wall components and secretory microbial molecules under in vitro and in vivo conditions. MAMPs could be used as potential molecular targets for treating bone-related diseases such as osteoporosis and periodontal diseases.

## 1. Introduction

The bone remodeling process is regulated by representative bone cells known as osteoclasts and osteoblasts [1]. The balance between bone-resorbing osteoclasts and bone-forming osteoblasts is essential for maintaining bone homeostasis [2]. However, imbalance between bone resorption and formation could lead to bone diseases [3]. Excessive osteoclast activity causes various bone diseases including osteoporosis, septic arthritis, osteomyelitis, and alveolar bone loss in periodontal diseases [4,5,6]. Especially, bacterial infections can directly affect bone homeostasis by increasing osteoclast differentiation and activation and/or decreasing osteoblast differentiation and activation [7]. For example, *Streptococcus pyogenes*, *Staphylococcus aureus*, and *Neisseria gonorrhoeae* are commonly found in patients with septic arthritis, resulting in cartilage and bone destruction within the joint [8]. *Staphylococcus* species such as *S. aureus* and *Staphylococcus epidermidis* are etiological agents of osteomyelitis [5]. Major oral pathogens, including *Porphyromonas gingivalis* and *Fusobacterium nucleatum*, are associated with periodontal diseases, manifesting alveolar bone loss [9]. However, unlike those pathogens, probiotics which are microorganisms that offer health benefits to the hosts are known to increase mineral density and volume of the bone [10]. For instance, *Lactobacillus reuteri* and *Lactobacillus rhamnosus* GG upregulate bone volume of mice [11,12]. In addition, other probiotics such as *Lactobacillus gasseri* and *Lactobacillus brevis* reduce bone loss and inflammation in mouse periodontitis model [13,14].

Bacteria have unique structural components called microbe-associated molecular patterns (MAMPs) including lipopolysaccharide (LPS), lipoteichoic acid (LTA), lipoprotein (LPP), and peptidoglycan (PGN) [15]. The recognition of MAMPs by pattern recognition receptors (PRRs) is crucial for inducing host immune responses [15]. In addition, secretory microbial molecules including short chain fatty acid (SCFA), extracellular vesicle (EV), extracellular polysaccharide, and cyclic dinucleotide (CDN) also modulate bone cells [16,17,18]. Therefore, for a clear understanding of the regulation of bone metabolism by bacteria, it is essential to understand the effects of MAMPs and secretory microbial molecules on bone cells and their regulatory mechanism. Based on those understanding, we could better prevent bacteria-mediated inflammatory bone diseases and formulate therapeutic strategies by using bacteria-derived substances.

## 2. Microbe-Associated Molecular Patterns

MAMPs are structural or secretory molecules that are highly conserved in most microbes [19]. Well-known MAMPs are bacterial polysaccharides (LPS and LTA), surface proteins (LPP and adhesin), PGNs, and secretory molecules (SCFA, EV, extracellular polysaccharide, and CDN) [20]. These MAMPs can be sensed by various host PRRs, such as Toll-like receptors (TLRs) and nucleotide-binding oligomerization domain (NOD)-like receptors (NLRs), or G-protein coupled receptors (GPCRs) [21,22]. Indeed, there are many host PRRs classified according to their location, function, and ligand specificity [23]. There are typically four types of PRRs: TLRs, NLRs, C-type lectin receptors, and RIG-1 like receptors [21]. Among these, TLRs localized at plasma membrane or in endosomes and NLRs localized in cytoplasm are the major PRRs in recognizing bacterial MAMPs [21]. For instance, TLR4 senses LPS, and TLR2 senses LPP and LTA [24]. On the other hand, NOD1 and NOD2 recognize bacterial PGNs through their distinct structural moieties, d-glutamyl-*meso*-diaminopimelic acid (iE-DAP) and muramyl dipeptide (MDP), respectively [25]. Based on their displayed patterns, each host receptor responds to its specific bacterial ligand, subsequently producing anti- or pro-inflammatory cytokines and chemokines to counteract against invading microbes [26]. It has been reported that pathogens or probiotics and their MAMPs could also affect osteoimmunological responses (Table 1) [27]. Therefore, we will focus on MAMPs and their effects on bone homeostasis in this section.

### 2.1. Bacterial Polysaccharides

#### 2.1.1. Lipopolysaccharide

LPS, also known as endotoxin, is a characteristic cell wall component of Gram-negative bacteria. It is composed of a hydrophobic lipid A, a hydrophilic core polysaccharide, and a hydrophilic O antigen-specific side polysaccharide chain (Figure 1) [58]. Lipid A, an anchoring part of LPS on bacterial outer membrane, plays a crucial role in inducing host immune responses [59]. O antigen is a sequential sugar molecule, which varies among bacterial species [59]. Bacteria could escape host immune responses like phagocytosis by using O antigen [59]. When Gram-negative bacteria infect the host, LPS is recognized by TLR4 in cooperation with other various host molecules, such as cluster of differentiation (CD) 14, LPS-binding protein, and myeloid differentiation (MD)-2 [60,61,62]. LPS-mediated TLR4 signaling is transferred through Toll/interleukin-1 receptor (TIR) domain-containing adaptor protein/myeloid differentiation factor 88 (MyD88) or TIR domain-containing adaptor inducing interferon (IFN)-β (TRIF)/TRIF-related adaptor molecule [63,64]. These downstream signals activate nuclear factor-κB (NF-κB) or IFN regulatory factor 3 (IRF3), prompting the production of pro-inflammatory cytokines such as interleukin (IL)-1β, IL-6, tumor necrosis factor (TNF)-α, nitric oxide, or type I IFNs [63,65,66].

To date, many reports have demonstrated that LPS induces bone loss at various sites in vivo [28,29,30,67]. For instance, Chen et al. showed that LPS reduces the number of trabecular bone and bone mineral density in mice [28]. Rid et al. reported that injection of LPS into gingival sulcus on rats triggers periodontal and alveolar bone damage by the induction of pro-inflammatory cytokines [29]. In addition, some reports demonstrated that LPS causes local bone resorption on murine calvaria [30,67]. Overall, numerous research indicates that LPS induces bone loss under various physiological conditions [28,29,30,67].

Unlike in vivo, LPS plays dual roles in osteoclastogenesis depending on the timing of LPS treatment during osteoclast differentiation in vitro [31,32]. LPS inhibits osteoclast differentiation when treated on the mouse bone marrow-derived macrophages also known as pre-osteoclasts, but it triggers osteoclastogenesis when treated on receptor activator of NF-κB (RANK) ligand (RANKL)-pretreated macrophages which are committed osteoclasts [32]. Mouse bone marrow-derived monocytes treated with macrophage-colony stimulating factor (M-CSF) and RANKL are differentiated into osteoclasts, while monocytes treated with M-CSF and LPS are not differentiated into osteoclasts, indicating that LPS cannot be substituted for RANKL [31]. In addition, LPS prevents RANKL-induced differentiation of mouse macrophages into mature osteoclasts [32]. In contrast, committed osteoclasts from mouse show different patterns from macrophages [32]. Retreatment with M-CSF and LPS to committed osteoclasts in the absence of RANKL leads to vigorous mature osteoclast differentiation, indicating that LPS is a potent osteoclastogenic factor in committed osteoclasts [32]. In consideration of bone loss effects by LPS in mouse or rat [28], we suggest that committed osteoclasts, rather than pre-osteoclasts, are more suitable to represent in vivo situation in bone.

Meanwhile, diminished osteoclastogenesis by TLR ligands seems to be a common phenomenon that occurs in macrophages [68]. When TLR ligands are treated, macrophages from mouse may preferentially cause to execute the host defense strategy rather than inducing osteoclast differentiation [68]. In fact, macrophages fail to differentiate into osteoclast and induce the pro-inflammatory cytokines when stimulated by TLR ligands including LPS, PGN, poly(I:C), CpG DNA, and LTA [31,32,68,69].

On the other hand, several studies demonstrated that osteoblast differentiation is suppressed by LPS through the downregulation of Runx2, osterix, and activating transcription factor (ATF) 4 expression [33,34,35]. In addition, LPS-stimulated osteoblasts induce osteoclastogenesis by producing pro-inflammatory mediators, such as IL-1, IL-6, prostaglandin E_2_ (PGE_2_), and RANKL, which are well-known osteoclast-activating factors [70,71,72,73]. In conclusion, LPS is a potent bone resorbing MAMP that upregulates osteoclastogenesis and downregulates osteoblastogenesis.

#### 2.1.2. Lipoteichoic Acid

LTA is one of the important virulence factors of Gram-positive bacteria, which consists of alditol phosphate-containing polymer and lipid anchor [74]. Based on its chemical structure, LTA is classified into five types (type I to V), and each bacterium has a distinct characteristic LTA structure (Figure 2) [74,75]. Bacterial LTA has association with various inflammatory diseases such as skin infection and sepsis [75]. LTA specifically attaches to the host cells through TLR2 and CD14, leading to the recruitment of MyD88 and TNF receptor associated factor (TRAF) 6 [76]. It sequentially induces mitogen-activated protein kinase (MAPK) and NF-κB activation [76]. Consequently, the downstream cascade induces innate immune responses, such as the production of nitric oxide and TNF-α [77,78].

In addition, LTA attenuates osteoclast differentiation from mouse bone marrow-derived macrophages [36,37,38]. LTA from *Enterococcus faecalis* inhibits differentiation of macrophages into mature osteoclasts. Macrophages seem to retain phagocytic activity against bacterial infection [36]. Recently, Wang et al. reported that these inhibitory effects occur through the transcription factor, recombination signal binding protein (RBP)-Jκ [37]. Not only *E. faecalis* LTA but also staphylococcal LTA inhibits osteoclastogenesis and bone resorption through TLR2 pathway [38].

LTA is also responsible for modulating osteoblast differentiation and bone formation. For instance, mesenchymal stem cells stimulated by LTA from *S. aureus* upregulate the expression of various osteogenic markers, such as Runx2, alkaline phosphatase (ALP), type I collagen, and calcium deposition through enhanced autophagy [39]. Additionally, LTA from *S. aureus* promotes the synthesis of bone bridge, ossification, and healing of femoral fractures induced by medial parapatellar arthrotomy [40]. It is likely that such phenomenon was induced by enhancing osteoblast differentiation and inhibiting osteoclast activation [40]. Although LTA appears to be a potential treatment for bone diseases, further studies are needed because LTA causes differential immune responses depending on the source of bacteria.

### 2.2. Surface Proteins

#### 2.2.1. Lipoprotein

Bacterial LPPs, which are anchored to the cell membrane by N-terminally-linked fatty acids, are one of the major virulence factors causing potent immuno-stimulatory effects [79]. The structure of LPP consists of a protein with a lipid moiety [80]. The protein is in charge of physiological functions, while the lipid moiety anchors LPPs in the bacterial cell membranes and induces bacteria-specific immune responses via TLR2 and additional receptors [80,81]. Based on the number of lipid moieties, LPPs are classified into diacylated or triacylated forms, which are mainly expressed on Gram-positive or Gram-negative bacteria, respectively [82]. Diacylated LPPs contain S-diacylated cysteine residues and triacylated LPPs have N-acyl-S-diacylated cysteine residues [82]. Meanwhile, TLR2 forms a heterodimer with TLR1 or TLR6, recognizing LPPs. Diacylated LPPs are sensed by TLR2/TLR6, while triacylated LPPs are recognized by TLR1/TLR2 (Figure 3) [83,84]. The recognition of LPPs through TLR1/TLR2 or TLR2/TLR6 heterodimer mediates MyD88-mediated signaling transduction and subsequently activates NF-κB, enabling the production of pro-inflammatory cytokines and chemokines [85].

In bone homeostasis, bacterial LPP is known as a potent bone-destructing factor. Kim et al. demonstrated that committed osteoclasts treated with wild-type *S. aureus* enhances osteoclast differentiation, whereas LPP-deficient *S. aureus* loses such effect [41]. Furthermore, synthetic lipopeptides, Pam2CSK4 and Pam3CSK4, which mimic bacterial LPP induce osteoclastogenesis by activating TLR2/MyD88 pathway and secreting pro-inflammatory cytokines such as IL-6 and TNF-α [41,42]. In addition, Pam2CSK4 and Pam3CSK4 upregulate RANKL production while downregulating osteoprotegerin (OPG) by stimulating osteoblasts [41].

LPPs also induce bone loss in vivo. Pam2CSK4 and Pam3CSK4 destruct calvarial bone in the mouse implanted with a collagen sheet [41]. In addition, intraperitoneal administration of Pam2CSK4 or Pam3CSK4 significantly decreases the femur bone density of mice [41,43]. Souza et al. also reported that Pam2CSK4 promotes periodontal destruction in mice by inducing gingival inflammation and alveolar bone resorption [43]. Consequently, bacterial LPPs induce differentiation of mouse committed osteoclasts into mature osteoclasts in vitro and bone resorption in vivo via TLR2/MyD88 pathway [41,42,43].

#### 2.2.2. Adhesin

Bacteria possess various macromolecules on their cell surface allowing adhesion and/or interaction with the host [86]. Therefore, these surface molecules play vital roles in bacterial pathogenesis and host immune responses. There are a number of carbohydrates and protein adhesins in both Gram-positive and Gram-negative bacteria [87]. Protein adhesins are further classified into fimbrial and non-fimbrial associated structures [87]. Among them, fimbriae are the most representative bacterial surface adhesins which stick out from the surface and play a major role in host cell invasion [88].

*Porphyromonas* fimbriae are known as a potent osteoclastogenesis factor. It was reported that fimbriae of *Porphyromonas gulae* and *P. gingivalis* induce osteoclast differentiation and cytokine production such as IL-1β, IL-6, and TNF-α in bone marrow-derived macrophages [44,45]. In addition, *P. gingivalis* fimbriae trigger bone resorption by utilizing tyrosine kinases [46,47]. *P. gingivalis* fimbriae affect osteoclast differentiation but not osteoblast differentiation [48]. Overall, bacterial fimbriae can induce bone resorption predominantly by inducing osteoclastogenesis in mice [44,45,46,47,48]. However, little is known about the effects of other adhesins on bone and bone-related cells. Therefore, it is essential to demonstrate how various bacterial adhesins affect bone remodeling in the days to come.

### 2.3. Peptidoglycan

PGN is a highly conserved bacterial cell wall component. It is made up of polymers composed of N-acetylglucosamines (NAGs) and N-acetylmuramic acids (NAMs). Each NAM has a short peptide chain that is involved in forming a cross-linked peptide bridge between polymers [89]. MDP (NAM-l-Ala-d-Glu) is a minimal essential structural motif of PGNs in both Gram-positive and Gram-negative bacteria (Figure 4) [89]. Gram-positive bacteria possess lysine-type PGNs, while Gram-negative bacteria have DAP-type PGNs [90]. 

NODs, which are present in the host cytoplasm, are responsible for sensing PGN motifs [91]. NODs are composed of caspase recruitment domain (CARD) at N-terminal, an NOD at intermediate site, and a leucine-rich repeat domain at C-terminal [91]. Among the NODs, NOD1 recognizes iE-DAP of Gram-negative bacterial PGNs, whereas NOD2 senses MDP moieties of ubiquitous bacterial PGNs [91]. Once iE-DAP and MDP are recognized by NOD1 and NOD2, respectively, both NODs induce CARD-CARD interaction and then form the complex with the adaptor molecules, receptor-interacting protein-like interacting caspase-like apoptosis regulatory protein kinase (RICK), leading to NF-κB and MAPK activation for triggering inflammatory responses (Figure 5) [91,92,93].

Accumulating reports suggested that PGN plays a bi-functional role in bone metabolism. Kishimoto et al. reported that PGN and LPS synergistically induce bone resorption and osteoclastogenesis [50]. They showed that *S. aureus* PGN or *Escherichia coli* PGN accelerates osteoclast formation and bone resorption only when they are co-stimulated with LPS. However, when treated independently, only *S. aureus* PGN, but not *E. coli* PGN, induces alveolar bone resorption [50]. Similarly, Ozaki et al. demonstrated that *S. aureus* PGN and *E. coli* LPS exacerbate alveolar bone loss and induce osteoclastogenesis from committed osteoclasts through the upregulation of TNF-α, IL-10, and IL-17 [51]. It was reported that MDP together with LPS, not MDP alone, could enhance osteoclastogenesis and bone loss through upregulation of RANKL and TLR4 expression [52]. PGN of *Actinomyces naeslundii* also induces osteoclastogenesis and alveolar bone resorption by triggering pro-inflammatory cytokines, such as IL-1β, IL-6, and TNF-α [53]. The importance of NOD1 stimulation has been suggested because stimulation of NOD1 induces alveolar bone loss and periodontitis [54]. In addition, Chaves et al. described that osteoclast formation is increased in NOD1 knockout mice, suggesting that NOD1 affects the upregulation of osteoclastogenesis [54]. Collectively, PGN-induced osteoclastogenesis and bone resorption occur through NOD1-related downstream signals and production of pro-inflammatory cytokines.

Unlike the NOD1 signaling, NOD2 signals could induce bone formation. Park et al. reported that MDP could enhance bone mineral density by upregulation of bone formation [57]. Osteoblasts treated with MDP augment Runx2 expression, which is a major transcriptional factor of osteoblast differentiation. In addition, MDP indirectly reduces osteoclastogenesis through the downregulation of RANKL/OPG ratio from osteoblasts. Furthermore, pre- or post-treatment of MDP alleviates RANKL-induced osteoporosis via NOD2 signaling [57]. Because MDP increases NOD2 expression level and other NOD2 ligands also induce bone formation similarly to the action of MDP, NOD2 agonists like MDP could be a novel therapeutic agent of osteoporosis [57]. *Lactobacillus fermentum*, which activates NOD2 signaling, attenuates bone resorption and decreases the number of osteoclasts [54]. Likewise, intraperitoneal injection of *L. plantarum* PGN in mice enhances bone density of femurs [57]. Collectively, NOD2-stimulating bacterial PGNs can ameliorate bone health by increasing bone formation and diminishing bone resorption.

### 2.4. Secretory Microbial Molecules

#### 2.4.1. Short Chain Fatty Acid

SCFAs, which consist of fewer than six carbons, are metabolites mainly produced by commensal bacteria through fermentation of dietary fibers [94]. Acetate, propionate, and butyrate are the most predominant form of SCFAs in the gastrointestinal tract and have a molar ratio varying from 40:40:20 to 75:15:10 depending on the diet [95,96]. There are two major pathways for modulation of host cells by SCFAs. One major pathway utilized by the host is through GPCRs. GPCRs are seven transmembrane receptors of host’s signaling molecules or MAMPs to induce intracellular signaling pathways [97]. Among GPCRs, SCFAs can bind and activate GPCR 40, 41, and 43, which are designated as free fatty acid receptor (FFAR) 1, 3, and 2, respectively, or GPCR 109a [98]. Conserved two arginine residues at transmembrane helixes 5 and 7 are important for the recognition of SCFAs by FFAR 1, 2, and 3 [99]. In addition, GPCRs are commonly expressed in bone metabolism-involved cells, including adipocytes, neutrophils, macrophages, osteoclasts, and osteoblasts [100,101,102,103]. Another route of host cell modulation by SCFAs is the inhibition of histone deacetylases (HDACs). HDAC plays an important role in regulating gene expression by epigenetic modification of chromosome structure [104]. Among SCFAs, butyrate inhibits HDAC activity, leading to decreased production of MAMP-induced nitric oxide or pro-inflammatory cytokines [105]. Consequently, SCFAs have various effects on the host’s health through the activation of GPCR or HDAC inhibition, such as modulating intestinal homeostasis, enhancing the production of antimicrobial peptides, providing anti-inflammatory immune responses, and augmenting mucosal vaccine properties [105,106]. SCFAs also affect bone metabolism by the regulation of osteoclasts and osteoblasts via GPCR or HDAC inhibition [107,108,109].

Iwami et al. reported that sodium butyrate potently attenuates the formation of tartrate-resistant acid phosphatase (TRAP)-positive multinucleated cells, which are cultured from bone marrow cells [16]. Besides, FFAR1 knockout mice show less bone density than wild-type mice. In addition, activation of FFAR1 suppresses the mRNA expression of osteoclast-specific genes, such as TRAP, matrix metalloproteinase-9 (MMP-9), and cathepsin K, and RANKL-induced osteoclastogenesis from macrophages via inhibiting RANKL-induced NF-κB signaling pathway [110]. Similarly, increased alveolar bone loss is observed in FFAR2 knockout mice. In fact, activation of FFAR2 by SCFAs or its agonists decreases the RANKL-induced osteoclast differentiation from macrophages and prevents alveolar bone loss [111]. On the other hand, mRNA expression levels of FFAR3 or GPCR 109a are downregulated during RANKL-induced osteoclast differentiation from macrophages. Moreover, inhibitory effect of SCFAs on osteoclastogenesis treated with RANKL is not observed if SCFAs were administered to committed osteoclasts [107]. Interestingly, two HDAC inhibitors, trichostatin A and sodium butyrate, inhibit the RANKL-induced osteoclast differentiation from macrophages through downregulation of osteoclast-specific gene expression, such as RANK and cathepsin K [108]. Because HDAC inhibition dampens osteoclast differentiation [103], inhibition of HDAC by SCFAs may be involved in the suppression of osteoclastogenesis [108]. Collectively, SCFAs inhibit the osteoclastogenesis from macrophages via HDAC inhibition and/or FFAR1 or 2 activation but not from committed osteoclasts [107,108,110,111].

SCFAs also influence osteoblast proliferation and differentiation in both animals and humans. Sodium butyrate increases the ALP activity of MC3T3-E1 murine osteoblastic cell line [16]. In contrast, a high concentration of sodium butyrate inhibits the differentiation and mineralization of ROS17/2.8 rat osteoblastic cell line via suppression of osteoblast-specific factors, such as Runx2, osterix, and Dlx5 [112]. Moreover, a low concentration of butyrate induces histone H3 acetylation with concurring expression of ALP, osteonectin, and OPG in MG-63 human osteoblastic cell line [109]. In fact, treatment of sodium butyrate at 16 mM attenuates osteoblast proliferation by suppression of cell cycle in vitro [113]. Notably, they have the potential to treat destructive bone diseases caused by postmenopausal or inflammatory conditions in animal models [114]. Thus, the optimal concentration of SCFAs could be used as therapeutic agents for treating bone diseases.

#### 2.4.2. Extracellular Vesicle

EVs could be released from archaea, eukaryote, and bacteria [115]. EVs are usually classified into three types (exosomes, microvesicles, and apoptotic bodies) according to their biogenesis [116]. Among them, the diameter of bacterial EVs is roughly 20~500 nanometers, and these spherical membrane-enveloped particles are secreted from parental bacteria into the extracellular environment [117]. Bacterial EVs carry diverse cargos such as membrane-bound proteins, LPPs, polysaccharides, enzymes, toxins, metabolites, and nucleic acids [118]. Bacteria can utilize EVs for horizontal gene transfer [119]. Various host cells recognize the content of EVs via diverse PRRs such as TLR, NOD, and retinoic acid-inducible gene, potentially leading to inflammatory conditions, or in some cases, immune tolerogenic conditions [120].

It has been reported that EVs derived from *Filifactor alocis* inhibit the differentiation of bone-derived mesenchymal stromal cells in vitro [17]. In addition, *F. alocis* EVs potently downregulate osteogenic factors, such as Runx2, osterix, ALP, osteocalcin (OCN), and type I collagen, thereby attenuating mineralization. Notably, *F. alocis* EVs activate TLR2 but not TLR4, and the inhibitory effect of *F. alocis* EVs on osteogenic differentiation is fully dependent on TLR2 signaling pathway which mediates the activation of MAPK and NF-κB [17]. Unfortunately, the role of bacterial EVs on osteoclast differentiation and function has been poorly investigated. However, emerging evidence indicates that bacterial EVs might directly or indirectly influence osteoclast differentiation. Bacterial EVs activate the monocyte-derived dendritic cells to induce pro-inflammatory cytokines, such as IL-1β, IL-6, or TNF-α, which can trigger the osteoclast differentiation from committed osteoclasts [121,122]. Moreover, bacterial EVs regulate the expression of RANKL and OPG, which are modulators of osteoclastogenesis, through TLR2 in mesenchymal stromal cells [17]. Further studies are needed to understand the role of bacterial EVs on bone metabolism.

#### 2.4.3. Extracellular Polysaccharide

Many bacteria produce extracellular polysaccharides which are classified into two types: exopolysaccharides (EPS) and capsular polysaccharides (CPS) [123]. EPS are defined as released polysaccharides around the bacterial cell surface, becoming an integral component of biofilm [124]. On the other hand, CPS are covalently bonded to the bacterial cell surface [125]. Bacteria exploit the extracellular polysaccharide as a barrier to protect themselves against harsh environments [126].

Mounting evidence suggested that extracellular polysaccharides of bacteria have bi-functional effects on bone metabolism. For example, EPS purified from *Bifidobacterium longum* (EPS-624) inhibit osteoclast differentiation from mouse bone marrow-derived macrophages by activating TLR2 signaling pathway [127]. In addition, EPS-624 increase the differentiation of osteoblasts from human bone marrow-derived mesenchymal stromal cells [127]. Thus, authors suggested the potential therapeutic use of EPS-624 against destructive bone diseases [127]. Moreover, EPS isolated from *Vibrio diabolicus*, which are hyaluronic acid-like EPS, potently enhance bone healing without abnormal bone growth in vivo [128]. Indeed, lying osteoblasts on trabecular bone surfaces and increasing osteocytes inclusion are observed in the bone treated with EPS [128]. In contrast, oversulfated EPS produced by *Alteromonas infernus* (OS-EPS) inhibit the proliferation and mineralization activity of osteoblasts in vitro [129]. Also, OS-EPS decrease the RANKL-induced osteoclast differentiation from CD14^+^ human monocytes while increasing the collagenolytic activity of osteoclasts by using cathepsin K [129]. Notably, OS-EPS cause trabecular bone loss through enhanced osteoclastogenesis [129].

Similar to EPS, CPS also affect the activation and differentiation of bone cells, including osteoclasts and osteoblasts. CPS from *Aggregatibacter actinomycetemcomitans* Y4 (Aa-CPS) enhance the formation of osteoclasts and promote the bone resorptive activity through the induction of IL-1α in vitro [130]. Aa-CPS also activate osteoclasts by upregulating PGE_2_, which has a positive effect on osteoclast formation in vitro [131]. These reports indicate that Aa-CPS-induced PGE_2_ and IL-1α are involved in inflammatory bone diseases such as periodontitis by promoting osteoclastogenesis. In addition, Aa-CPS show anti-proliferative activity by causing Fas-mediated apoptotic cell death in MC3T3-E1 murine osteoblastic cell line in vitro [132]. Moreover, immunization of CPS from *P. gingivalis* exhibits immunoglobulin responses and protects *P. gingivalis-*induced oral bone loss in vivo, suggesting that CPS are one of the responsible molecules for bone diseases [133]. In conclusion, because extracellular polysaccharides have controversial effects on the bone metabolism, further studies are needed.

#### 2.4.4. Cyclic Dinucleotide

CDNs were originally identified in 1987 as bacterial second messengers that regulate cellulose synthesis [134]. Bacterial CDNs are classified as cyclic diadenylate monophosphate (c-di-AMP), cyclic diguanylate monophosphate (c-di-GMP), and 3′,3′-cyclic guanosine monophosphate-adenosine monophosphate (3′3′-cGAMP) [135]. CDN contains two nucleotide monophosphates that are linked to each other by phosphodiester bonds to form a cyclic structure [135]. CDNs are important for the maintenance of bacterial life cycle including survival, colonization, and biofilm formation [136]. For instance, CDNs trigger extracellular matrix production, subsequently forming biofilm in bacteria [137]. Furthermore, CDNs released from bacteria can be recognized by host cells and activate various host immune responses [138].

Stimulator of IFN genes (STING), also known as transmembrane 173, has 4 transmembrane regions and is located at the endoplasmic reticulum of host cells [139]. STING directly recognizes the cytosolic CDNs, leading to secretion of type I IFNs (Figure 6) [140]. When CDNs bind to STING, STING recruits TRAF-associated NF-κB activator-binding kinase 1 (TBK1) to the C-terminal tail [141]. Recruited TBK1 phosphorylates IRF3, forming a homodimer [142]. Phosphorylated IRF3 homodimer enters through the nucleus pore, inducing the gene expression of IFN-β [142]. Several studies investigated that STING is involved in bone metabolism. Overexpression of STING in RAW 264.7 cell inhibits RANKL-induced osteoclast differentiation and expression of osteoclast-specific genes, such as TRAP, cathepsin K, and MMP-9 [143]. On the other hand, lack of STING suppresses bone accrual via inhibition of pro-osteogenic gene expression [144].

Interestingly, CDNs can also influence bone metabolism. CDNs inhibit RANKL-induced osteoclast differentiation from mouse bone marrow-derived macrophages through STING-dependent signaling pathway [18]. Authors demonstrated that CDNs potently trigger STING-TBK1-IRF3 cascade and induce the mRNA expression of IFN-β during osteoclast differentiation from macrophages. IFN-β in turn acts as a negative regulator of osteoclast differentiation by activating Janus kinase (Jak)-signal transducer and activator of transcription (STAT) signaling [145], which is the major pathway responsible for the inhibition of osteoclastogenesis. In contrast, because ubiquitin-mediated degradation of Jak in committed osteoclasts, CDN-induced IFN-β cannot activate the Jak-STAT signaling pathway during differentiation of committed osteoclasts from mouse. Notably, CDNs prevent the RANKL-induced bone destruction in collagen sheet implanted mouse model [18]. Recently, it has been reported that STING interacts with TRAF6, which is a TLR signaling mediator, in human keratinocytes and monocytes [146]. Thus, further studies are needed to understand cooperative effects of MAMPs and CDNs on bone metabolism. In conclusion, targeting STING using CDNs serves as a novel therapeutic strategy for bone disorder treatment due to its heavy regulatory association with bone metabolism.

## 3. Therapeutics

Microbes influence bone metabolism by constant interaction with host using their various MAMPs (Table 2) [7]. In infectious condition, MAMPs often trigger immoderate osteoclastogenesis or inhibit osteoblast differentiation through the activation of immune responses, causing bone diseases such as osteomyelitis, osteoporosis, and periodontitis [7]. Antibiotics are commonly used to treat MAMP-induced bone diseases in bacterial infection [147]. Nevertheless, the emergence of antibiotic-resistant bacteria and remaining MAMPs after treatment pose significant challenge for complete clearance [148]. Therefore, further studies are needed to understand the role of MAMPs in bone diseases and to control the immune responses induced by MAMPs.

On the other hand, several studies investigated that some MAMPs, especially derived from probiotics, decrease bone resorption or enhance bone formation by controlling the differentiation of osteoclasts or osteoblasts, respectively, in both in vitro and in vivo studies [18,57,114]. Many therapeutic drugs, such as bisphosphonates, monoclonal antibodies, or hormone preparations, are traditionally developed to treat bone diseases by inhibiting bone resorption or inducing bone formation [149,150,151]. However, conventional drugs show unexpected side effects, such as nausea or osteonecrosis of jaw [151,152,153]. Therefore, we suggest that probiotic-derived MAMPs could alternatively be used in place of conventional therapies. To evaluate their therapeutic use, we have discussed below how to treat MAMP-induced bone diseases and how to exploit MAMPs in bone health.

### 3.1. Treatment of Microbe-Associated Molecular Patterns-Induced Bone Diseases

In general, most MAMPs are potent inducers of pro-inflammatory cytokines, such as IL-1, IL-6, or TNF-α, via the recognition by PRRs on animal and human cells, including epithelial cells, endothelial cells, and immune cells [154]. These MAMP-induced pro-inflammatory cytokines positively influence the differentiation of animal and human committed osteoclasts into mature osteoclasts and the activity of osteoclasts, leading to bone loss [155]. Thus, targeting pro-inflammatory cytokines or their receptors can become one of the therapeutic strategies for MAMP-induced bone diseases. For example, through the use of blocking antibodies that antagonize TNF-α and IL-1 receptors, significant reduction of inflammation and osteoclastogenesis were observed in experiment using mouse committed osteoclasts in vitro [41]. Antibody specific to IL-6 or IL-6 receptor directly inhibits osteoclast differentiation from mouse committed osteoclasts in in vitro experiment and restores bone erosion in TNF-α-transgenic mice [41,156]. Furthermore, many preclinical and clinical studies reported that TLR inhibitors or blocking antibodies alleviate inflammatory diseases [157]. Since MAMPs directly promote osteoclast differentiation from mouse committed osteoclasts through the activation of PRR signaling pathway in both in vitro and in vivo, regulation of MAMPs using inhibitors or antibodies may also be effective to control MAMP-induced osteoclastogenesis [41,158].

In the case of osteoblasts, targeting and inhibiting some potent MAMPs involved in diminishing osteoblasts could be a useful way to alleviate bone diseases [159]. For instance, hindering the action of LPS, which is a potent osteoblast inhibitor, might be valuable to prevent LPS-induced bone loss in bacterial infections [159,160,161,162]. There are several ways to inhibit the action of LPS. Jung et al. demonstrated that TLR4 decoy receptor inhibits LPS-induced NF-κB activation in human lymphatic microvascular endothelial cells in vitro and prevents Gram-negative bacterial sepsis in LPS-induced sepsis mouse model [159]. Indeed, anti-TLR4-antibody effectively treated stroke in vivo, hinting the therapeutic potency of anti-TLR4-antibody [160]. Therefore, blocking TLR4 by its specific antibody or decoy receptor might be helpful in LPS-induced osteoporosis patients. Another way to prevent LPS from binding osteoblasts is direct neutralization of the LPS. There are several peptides that can bind and inhibit LPS-induced inflammation. Antitoxin peptide Pep 19-2.5, which is designed to bind to LPS, reduced TNF-α expression and inflammation in several in vitro and in vivo models [161]. In addition, polymyxin B, which neutralizes LPS, shut down NF-κB signaling pathway in vitro [162]. As mentioned, LPS-neutralizing peptides could inhibit downstream signaling pathway of LPS and, therefore, are expected to be useful in preventing LPS-induced osteoblast reduction. Furthermore, LPPs or adhesins, which negatively affect osteoblast differentiation, could also be prevented by blocking osteoblast recognition receptors or neutralizing the MAMPs by antibodies or neutralizing peptides.

### 3.2. Probiotics as Therapeutic Agent for Bone Health

Recent decades, numerous studies are implemented to understand the role of commensal microbiota on digestive, endocrine, nervous, and immune system in the host [163]. Notably, emerging evidence indicates that commensal microbiota could regulate bone metabolism by controlling immune function and enhancing barrier function (Table 3) [11,164].

For instance, supplementation of probiotics, microorganisms that offer health benefit to the host, prevents bone loss in postmenopausal ovariectomized mice [11,164]. In addition, *L. reuteri* ATCC 6475 upregulates bone volume/tissue volume (BV/TV), trabecular number (Tb.N), and trabecular thickness (Tb.Th) in normal mice, postmenopausal ovariectomized female mice, and type I diabetic bone loss male mice [12,165,166]. *L. rhamnosus* GG increases BV/TV and OCN level but decreases RANKL, TNF-α, and IL-17 mRNA expression in ovariectomized osteoporotic mice [11]. *Lactobacillus paracasei* and *L. plantarum* enhance BV/TV, Tb.N, and cortical bone in osteoporotic mice [168]. *L. casei* prevents wear-debris induced osteolysis [169]. Furthermore, *L. rhamnosus* GG, *L. gasseri* SBT2055, or *L. brevis* CD2 downregulates alveolar bone loss and inflammation in periodontitis animal model [13,14,167]. These reports suggest that probiotics are enough to induce bone formation and reduce bone resorption in the host, presenting their possibility as novel bone healing agents. Moreover, the underlying action mechanisms of probiotics on bone metabolism are being elucidated. A well-known mechanism is that probiotic-derived MAMPs, such as cell wall components and secretory molecules, interact with bone cells, resulting in the regulation of bone metabolism.

As mentioned previously, MDP has positive effects on bone metabolism. MDP potently augments osteoblast differentiation and induces bone formation through the activation of NOD2 signaling pathway [57]. MDP has both preventive and therapeutic effects against RANKL-induced osteoporosis mouse model [57]. Other NOD2 ligands, including M-TriLYS, L18-MDP, and murabutide, also promote osteoblast differentiation in vitro and upregulate bone volume when injected in normal mice [57]. Furthermore, MDP indirectly inhibits osteoclastogenesis with attenuated RANKL/OPG ratio [57]. Notably, increased RANKL/OPG ratio can be observed in postmenopausal women with low bone mineral density [170], suggesting that MDP can effectively be used to treat patients with abnormal RANKL/OPG ratio. Collectively, MDP and other NOD2 agonists including NOD2-stimulating PGNs are promising therapeutic agents for the prevention or treatment of bone diseases by controlling the differentiation of both osteoclasts and osteoblasts.

Probiotic-derived secretory molecules, such as SCFAs and CDNs, also influence bone metabolism. SCFAs directly decrease the formation and function of mouse osteoclasts by HDAC inhibition and change the metabolic condition of mouse bone marrow cells in vitro [114]. In addition, adequate dose of SCFAs upregulates ALP activity of MC3T3-E1 mouse osteoblastic cell line in vitro [16]. Low dose of SCFAs increases osteoblast differentiation by HDAC inhibition in vitro using MG-63 human osteoblastic cell line [109]. Moreover, SCFAs indirectly increase osteoblast differentiation in wild-type mice compared to T-cell receptor knockout mice in which butyrate-activated T cells release the osteoblast differentiation factor, Wnt10b [171,172]. Furthermore, SCFAs systemically increase bone density in postmenopausal ovariectomized or collagen-induced inflammatory arthritis mouse model [114]. Like SCFAs, CDNs which are secretory bacterial second messengers also decrease osteoclast differentiation from mouse macrophages in vitro via STING-mediated IFN-β signaling pathway. In addition, CDNs potently prevent calvarial bone loss in collagen sheet implanted mouse model [18]. It is likely that bacterial secretory molecules, such as SCFAs and CDNs, can be developed as therapeutic agents for the treatment of bone diseases by inhibiting excessive osteoclastogenesis and promoting the activity of osteoblasts. Further studies are necessary to fully understand the regulatory effects of probiotic-derived molecules on osteoclast or osteoblast differentiation and activation.

## 4. Conclusions

In this review, we discussed the properties of MAMPs on bone metabolism in both in vitro and in vivo conditions. The MAMPs, including the cell wall components and secretory molecules, can directly or indirectly modulate the differentiation and activation of osteoclasts and osteoblasts. Thus, the MAMPs could be promising molecular targets for bacteria-induced bone diseases such as osteoporosis and periodontal diseases. Furthermore, a use of beneficial MAMPs on bone metabolism might be a novel therapeutic strategy for prevention or treatment of bone disorders.

## Figures and Tables

**Figure 1 ijms-22-05805-f001:**
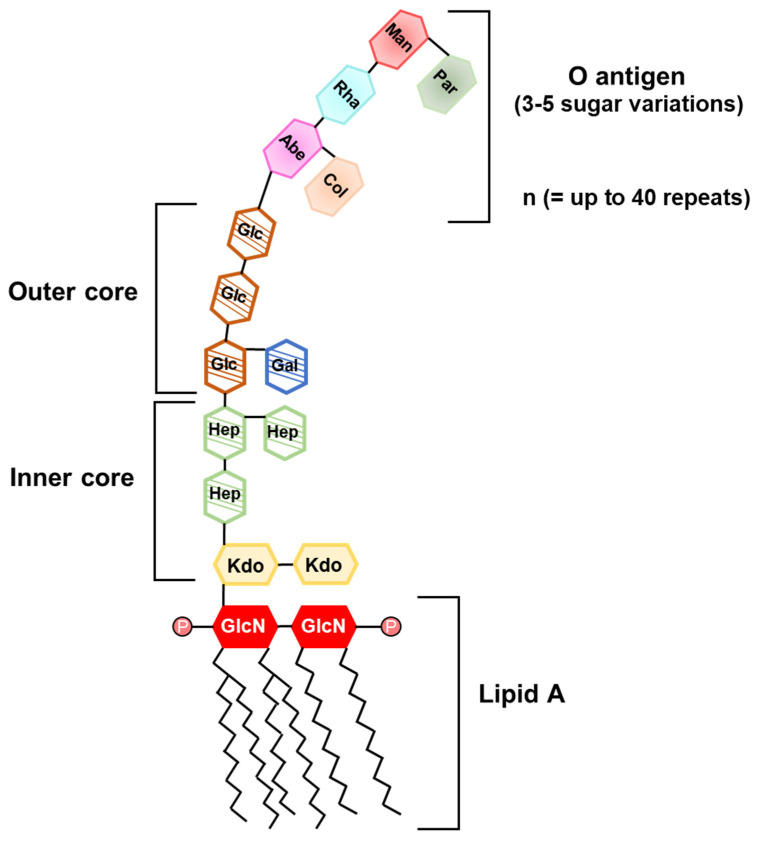
Illustration of lipopolysaccharide (LPS) structure. LPS is a potent immuno-stimulatory molecule of Gram-negative bacteria. It is composed of O antigen, outer or inner core polysaccharide, and lipid A. O antigen consists of repeating sugar molecules (n can be up to 40 repeats) and outer or inner core is a continuous polysaccharide chain. Composition and length of the O antigen and core polysaccharide are varied among bacterial strains. Lipid A consists of two phosphorylated glucosamines and acyl chains. The number of acyl chains and branched points in lipid A vary among bacterial species. Man, Mannose; Par, Paratose; Rha, Rhamnose; Abe, Abequose; Col, Colitose; Glc, Glucose; Gal, Galactose; Hep, Heptose; Kdo, 3-deoxy-d-manno-2-octulosonic acid; GlcN, Glucosamine; P, phosphate.

**Figure 2 ijms-22-05805-f002:**
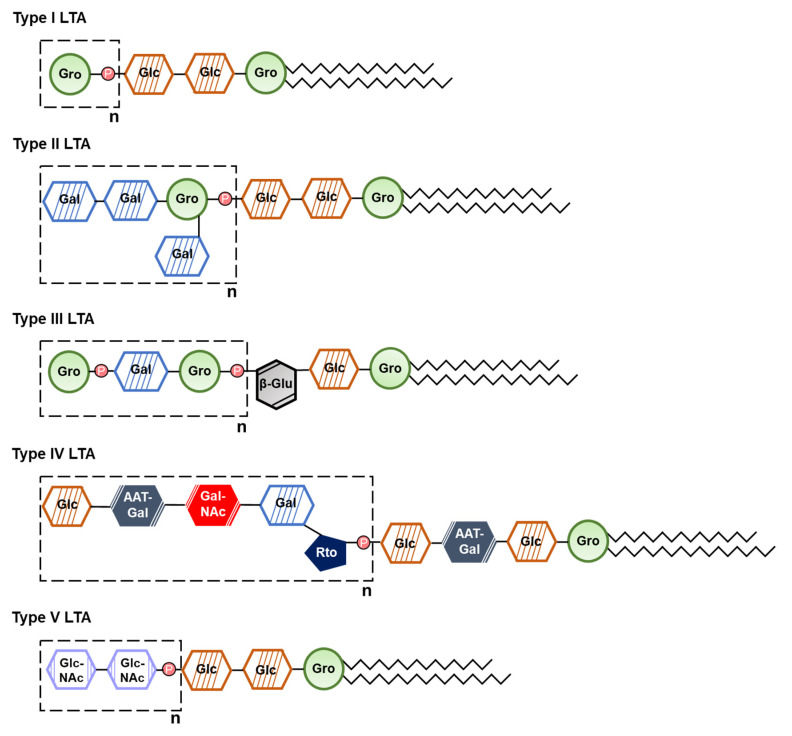
Illustration of lipoteichoic acid (LTA) structure. LTA is a Gram-positive bacterial cell wall component which is responsible for stimulating immune responses of hosts. There are five types of LTA which varies among bacterial species. Gro, Glycerol; Glc, Glucose; Gal, Galactose; β-Glu, β-Glucan; AAT-Gal, 2-acetamido-4-amino-2,4,6-trideoxy-d-galactose; Gal-NAc, N-acetylgalactosamine; Rto, Ribitol; Glc-NAc, N-acetylglucosamine.

**Figure 3 ijms-22-05805-f003:**
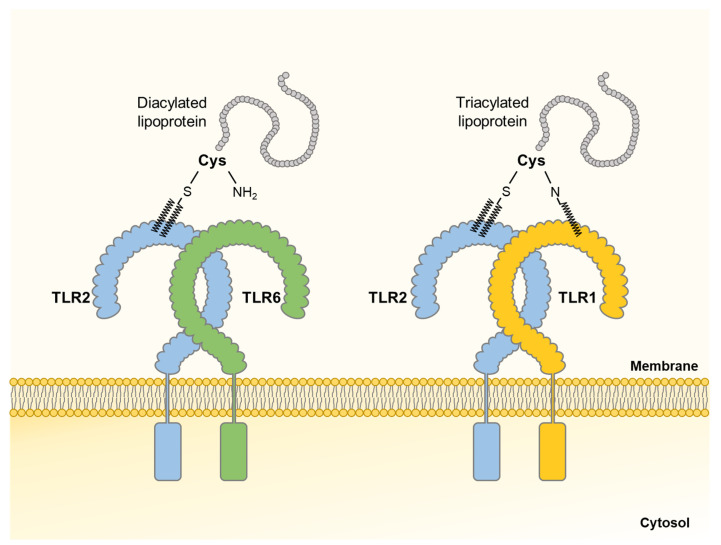
Structures of Toll-like receptor (TLR) 2 heterodimers and lipoproteins (LPPs). TLRs consist of extracellular leucine-rich repeat, transmembrane helix, and intracellular Toll/interleukin-1 receptor domain. Bacterial LPPs bind to the extracellular domains of TLR. Especially, TLR2/TLR6 heterodimers recognize diacylated LPPs and TLR1/TLR2 heterodimers sense triacylated LPPs.

**Figure 4 ijms-22-05805-f004:**
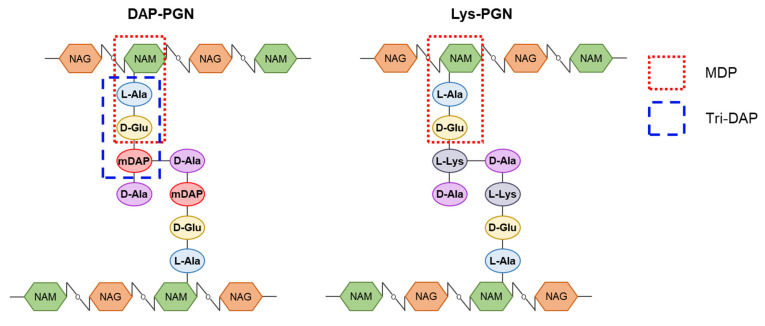
Illustration of peptidoglycan (PGN) structure. PGN is composed of two amino sugars, N-acetylmuramic acid (NAM) and N-acetylglucosamine (NAG), and amino acids. NAM and NAG are connected by β-1,4-glycosidic linkage. The peptide chain of three to five amino acids attaches to NAM. Lysine-type PGN and diaminopimelic acid (DAP)-type PGN contain a lysine and a *meso*-DAP at the third position of the peptide stem and predominantly found in Gram-positive and Gram-negative bacteria, respectively.

**Figure 5 ijms-22-05805-f005:**
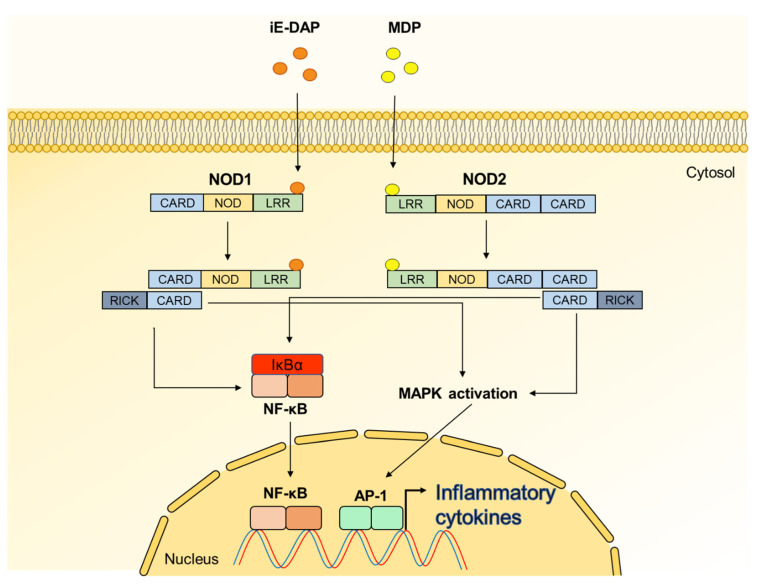
Signaling pathway of nucleotide-binding oligomerization domain (NOD) 1 and NOD2. d-glutamyl-*meso*-diaminopimelic acid (iE-DAP) of Gram-negative bacteria and muramyl dipeptide (MDP) of both Gram-positive and Gram-negative bacteria are recognized by NOD1 and NOD2, respectively. After activation of NODs, RICK is recruited through CARD-CARD interactions, leading to the activation of NF-κB and MAPK pathways.

**Figure 6 ijms-22-05805-f006:**
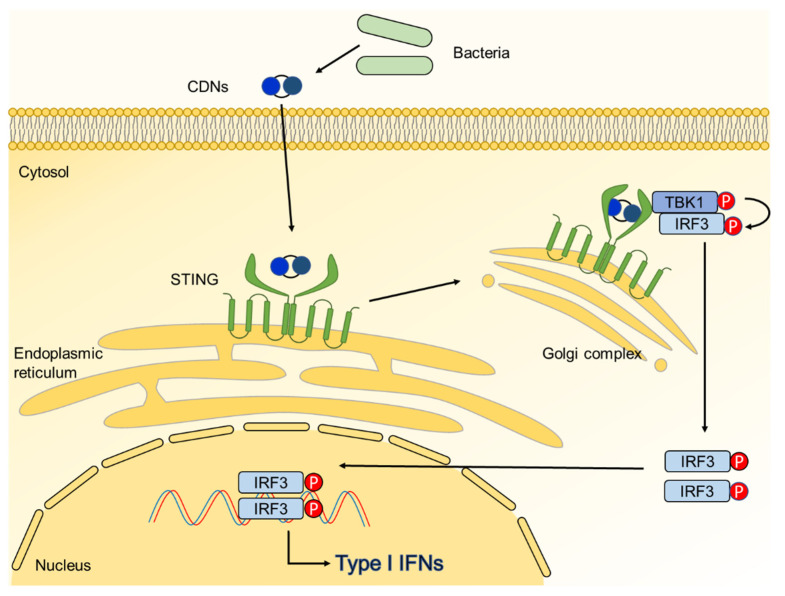
Signaling pathway of stimulator of interferon genes (STING). Bacterial cyclic dinucleotides (CDNs) are recognized by STING localized at endoplasmic reticulum. When STING is activated by CDNs, it is translocated from endoplasmic reticulum to the Golgi complex. This translocation triggers the STING-TBK1-IRF3 signaling cascade to increase expression of type I interferons.

**Table 1 ijms-22-05805-t001:** Effects of cell wall components on bone metabolism.

MAMPs	Receptor	Effects	References
Lipopolysaccharide	TLR4	Inducing bone loss Inhibiting osteoclastogenesis on macrophages Facilitating osteoclast differentiation on committed osteoclasts Downregulating osteoblast differentiation	[28,29,30,31,32,33,34,35]
Lipoteichoic acid	TLR2	Healing femoral fractures in mice Attenuating osteoclast differentiation and activating phagocytosis Upregulating osteogenic markers and osteoblastogenesis	[36,37,38,39,40]
Lipoprotein	TLR2	Promoting bone resorption Upregulating osteoclast differentiation Stimulating osteoblasts to elevate RANKL/OPG ratio	[41,42,43]
Fimbria	TLR4	Inducing osteoclastogenesis and bone resorption	[44,45,46,47,48,49]
Peptidoglycan	NOD1	Enhancing osteoclastogenesis and bone resorption Triggering osteoclast differentiation synergistically with LPS	[50,51,52,53,54,55,56]
	NOD2	Upregulation of bone density Facilitating osteoblast differentiation Diminishing osteoclastogenesis by reducing RANKL/OPG ratio	[54,57]

**Table 2 ijms-22-05805-t002:** Effects of secretory microbial molecules on bone metabolism.

MAMPs	Mechanism	Effects on Bone Metabolism	References
Short chain fatty acids	Activation of GPCRs Inhibition of histone deacetylases	Inhibited osteoclast differentiation and function Upregulated osteogenic factors in low dose Attenuated osteoblast differentiation and mineralization Prevented bone loss in various mouse models	[109,111,112,114]
Extracellular vesicles	Activation of TLR2 Induction of pro-inflammatory cytokines	Downregulated osteoblast differentiation and activity Regulated RANKL and OPG expression in mesenchymal cells	[17]
Extracellular polysaccharides	Activation of TLR2	Inhibited osteoclast differentiation from macrophages, but some EPS increased collagenolytic activity of osteoclasts Enhanced osteoblast differentiation, but oral pathogen-derived CPS decreased proliferation of osteoblasts	[127,128,129,132]
Cyclic dinucleotides	Induction of STING-mediated IFN-β	Inhibited differentiation of macropahges into mature osteoclasts Alleviated RANKL-induced bone destruction	[18]

**Table 3 ijms-22-05805-t003:** Benefits of probiotics on bone health.

Probiotics	Bone Effects		Animal Model	References
	Increase	Decrease		
*L. reuteri* ATCC 6475	BV/TV, Tb.N, Tb.Th, OCN		Normal	[12]
	BV/TV, Tb.N, Tb.Th	RANKL, TRAP5	Ovariectomy	[165]
	BV/TV, OCN, Wnt10b		Diabetic osteoporosis	[166]
*L. rhamnosus* GG	BV/TV, OCN	RANKL, TNF-α, IL-17	Ovariectomy	[11]
		Bone loss, Inflammation	Periodontitis	[167]
*L. paracasei* and *L. plantarum*	BV/TV, Tb.N, Cortical bone		Ovariectomy	[168]
*L. casei*		Osteolysis	Calvarial resorption	[169]
*L. gasseri* SBT2055		Bone loss, Inflammation	Periodontitis	[13]
*L. brevis* CD2		Bone loss, Inflammation	Periodontitis	[14]

## Data Availability

Not applicable.

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
