# Peer review of "Regulation of Bone Cell Differentiation and Activation by Microbe-Associated Molecular Patterns"

_ijms, 2021, doi:10.3390/ijms22115805_

Round 1

Reviewer 1 Report

ijms-1214208-peer-review-v1

General Comments: The authors have done a very nice job presenting and writing this thorough review. Almost every time I had a critique or addition, it was covered in the next sentence. Figures and tables are superb. My comments are minor, but are listed below:

  • The paper needs mild English proofing/editing.
  • In 1.1, hematopoiesis is also a key function of bone as a tissue.
  • Line 214 – Change “in vivo situation” to “in vivo situation in bone”
  • Line 400 – In the heading, write out “EV”
  • Line 506 – Osteosarcoma has been shown to not truly be a side effect of PTH. Therefore, this should be removed.

Reviewer 2 Report

General comments:

I) The review contains several paragraphs on osteoblasts, osteoclasts and osteocytes that are already reviewed which shall be deleted.

II) There was a lack of chemical structures of the secreted molecules as well as molecular mechanisms on the interactions between secreted molecules from bacteria to the receptors of the hosts. There was no detailed structural information on active sites of the receptors.

III) A few schemes shall be added to illustrate cell signaling.

IV) Several statements were insufficiently supported by references.

V) Several cellular or animal models were insufficiently specified. Sometimes there were mixed in vivo and in vitro findings which make their extrapolation or their possible clinical applications hazardous. A clear delineation between cellular, animal, human clinical findings shall be performed.

VI) Table 1, Table 2 and Table 3 were never cited and commented in the whole text.

VI) Conclusion was too broad.

Minor comments:

1) Abstract, lines 15-16, p1: Delete “Bone homeostasis is maintained by the balance between bone-resorbing osteoclast and bone-forming osteoblast through the regulatory action of hormones, cytokines and chemokines.”

2) Abstract, lines 17-19, p1: Specify receptors in “In particular, 17 the modulation of innate immunity and bone homeostasis is mediated through the interaction be- 18 tween microbe-associated molecular patterns (MAMPs) and the host pattern recognition receptors.”

3) Abstract, lines 20-21, p1: Specify pathogen bacteria in “Pathogenic bacteria tend to induce bone destruction and cause various inflammatory bone diseases including periodontal diseases, osteomyelitis, and septic arthritis.”

4) Abstract, lines 24-26, p1: Add specific findings to support adequately the scope of the review in “This review focuses on the regulation of osteoclast and osteoblast by MAMPs and secretory microbial molecules illuminating the proof-of-concept studies for treating various bone diseases.”

5) Bone homeostasis, from line 30, p1 to line 65 p2: The paragraph concerning bone homeostasis was already reported in many reviews. Delete Figure 1 and “Bone is considered a dynamic organ that undergoes constant remodeling throughout our whole life [1]. Remodeling process causes a replacement of the old bone with a new one, providing continuous mechanical support and vital protection for various organs 34 such as brain, heart and bone marrow [2]. In addition, bone plays a crucial role in calcium and phosphate homeostasis [3]. Therefore, meticulous regulation of bone remodeling and homeostasis is important for maintaining overall health of the human. The bone remodeling process is regulated by representative bone cells known as osteoclasts and osteoblasts [4]. The balance between bone-resorbing osteoclasts and bone-forming osteoblasts is an essential process to maintain bone homeostasis [5]. As shown in Figure 1, bone remodeling consists of four phases: bone resorption, reversal, bone formation, and mineralization [6]. In the bone resorption phase, osteoclasts recruited by osteocytes form sealing zones and generate ruffled borders where vacuolar H+-adenosine triphosphatase transport protons to the resorption compartment for acidification [7,8]. Organic components of the bone matrix are degraded by osteoclast-derived lysosomal proteolytic enzymes such as cathepsin K in the resorptive area [9]. In addition, tartrate-resistant acid phosphatase (TRAP) and matrix metalloproteinase-9 (MMP-9) are also secreted from osteoclasts to decompose the bone matrix [10]. After the resorption phase, some osteoclasts undergo programmed cell death to maintain the balance in bone remodeling [11]. During the reversal phase, bone resorption converts into bone formation. Zhao et al. suggested that ephrinB2 on mature osteoclasts binds to ephrinB4 on osteoblasts which transduces bidirectional signals resulting in the conversion into bone formation phase [12]. The binding interaction leads to the maturation of osteoblasts that synthesize a new bone matrix through the secretion of collagen-rich osteoid matrix and the regulation of osteoid mineralization. [13]. At the end of the bone-forming phase, osteoblasts can either be differentiated into osteocytes and bone-lining cells or undergo apoptosis, but their precise mechanisms remain elusive [14,15].”

6) Osteoclast, from line 67, p2 to line 100 p2: The paragraph concerning osteoclasts was already reported in many reviews. Delete Figure 2 and “Osteoclasts are representative skeletal cells that induce bone resorption. They are differentiated from monocyte/macrophage lineage of hematopoietic stem cells by two key osteoclastogenesis factors, macrophage-colony stimulating factor (M-CSF) and receptor activator of nuclear factor κB ligand (RANKL) [16]. Especially, M-CSF induces proliferation, differentiation, and survival of osteoclast precursors by binding macrophage colony- stimulating factor 1 receptor (c-Fms) [17]. Once RANKL engages with receptor activator of nuclear factor κB (RANK), osteoclast differentiation and activation are facilitated [18]. Binding of RANKL to RANK triggers recruitment of tumor necrosis factor (TNF) receptor-associated factor 6 (TRAF6) to the cytoplasmic domain of RANK and sequentially activates downstream signals like mitogen-activated protein kinase (MAPK), nuclear factor κB (NF-κB), c-Fos, and nuclear factor of activated T cells 1 (NFATc1) [18,19]. Through the activation, macrophages, also known as pre-osteoclasts, are differentiated into committed osteoclasts. Committed osteoclasts are the cells whose fates are determined to differentiate into osteoclasts with weak TRAP activity that exist in the mononucleated form [20]. Committed osteoclasts are further differentiated by M-CSF and RANKL and induce the expression of TRAP and dendritic cell-specific transmembrane protein (DC-STAMP), forming multinucleated and activated osteoclasts [16]. As shown in Figure 2, mature osteoclasts tightly attach to the organic matrix of bone surface using their integrin and induce bone resorption by utilizing osteoclast-specific molecules such as TRAP, cathepsin K, and MMP-9 [16,21]. On the other hand, osteoblasts and osteocytes secrete osteoprotegerin (OPG) to control the bone resorption. OPG is a decoy receptor of RANKL in which OPG forming the complex with RANKL impedes RANKL association with RANK on osteoclasts, hence inhibiting osteoclast differentiation [22]. Therefore, RANKL/OPG ratio is animportant indicator of regulating osteoclast differentiation. Collectively, those sophisticated adjustments of osteoclast differentiation are involved in the bone remodeling process, especially in bone resorption.”

7) “Osteoblast, from line 101 p3 to line line 129, p4: The paragraph concerning osteoblasts was already reported in many reviews. Delete Figure 3 and “Osteoblasts, derived from mesenchymal progenitors, are involved in bone formation, 1deposition, and mineralization [23]. Mesenchymal progenitors could be differentiated into not only osteoblasts but also myoblasts, chondrocytes, or adipocytes [24]. A major 1transcription factor, runt-related transcription factor 2 (Runx2), prevents mesenchymal stem cells from differentiating into the chondrocytic or adipocytic lineages but rather designates them into the osteoblastic lineage [25]. After differentiating into pre-osteoblasts, they become immature osteoblasts by the action of Runx2, osterix, and β-catenin and express high level of osteopontin (OPN) [25]. Immature osteoblasts can release bone matrix proteins, inhibiting the differentiating potential into chondrocytic lineage [26]. Sequentially, immature osteoblasts differentiate into mature osteoblasts expressing high levels of osteocalcin (OCN) [27]. Finally, they become osteocytes embedded in the bone matrix [28]. Osteoblast-related transcription factors, such as Runx2, osterix, and β-catenin, are activated by the development signals including hedgehog, notch, Wnt, bone morphogenetic protein, and fibroblast growth factor signalings [29-33]. Osteoblasts activated by transcription factors, such as Runx2, osterix, and β–catenin, express alkaline phosphatase (ALP), collagen type I α1 (COL1A1), OPN, and OCN [34]. ALP stabilizes the bone matrix in the early osteogenic phase [35]. COL1A1 is also an early marker of osteoblastogenesis, which consists the bone matrix [36]. OPN and OCN compose non-collagenous bone matrix, which are relatively expressed at late differentiation stage of osteoblasts [36]. In conclusion, osteoblasts are differentiated and activated, and thereby being involved in bone formation processes (Figure 3).”

8) Osteocyte, lines 130-141, p4: The paragraph concerning osteoblasts was already reported in many reviews. Delete “Osteocytes derived from osteoblasts are located in mineralized bone matrix [37]. When bone matrix is abundant, osteoblasts become osteocytes or bone-lining cells which integrates into the matrix [38]. One of many functions of osteocyte is communication with other neighboring osteocytes and bone cells in canaliculi [37]. Osteocytes connect osteoblast and bone-lining cells through gap junctions which facilitate intracellular transportation of small signaling molecules [39]. In addition, the osteocyte networks are mechanosensors detecting mechanical stresses [40]. Mechanical loading stimulates osteocytes toproduce prostaglandin E2 (PGE2), prostacyclin, nitric oxide, and insulin-like growth factor-1 to upregulate osteoblast differentiation, resulting in increased bone matrix [41-43]. On the other hand, sclerostin, which inhibits osteoblast activity by binding to several osteoblastogenic factors such as BMP, ALP, and COL1A1, is also produced by osteocytes in both loading and unloading situation [44,45].”

9) Inflammatory bone diseases, lines 153-154, p4: Specify types of probiotics and add explanations to support “However, unlike those pathogens, probiotics are known to 154 increase mineral density and volume of the bone [53].”

10) Inflammatory bone diseases, from line 159, p4 to line 161, p5: Add references to support “In addition, secretory microbial molecules including short chain fatty acids (SCFAs), extracellular vesicle (EV), extracellular polysaccharide and cyclic dinucleotide (CDN) also modulate bone cells.”

11) MAMPs, title, line 167 p5: Replace MAMPs by its full name in the title.

12) MAMPs, lines 168-169, p5: Specify types of polysaccharides, surface proteins, PGNs, and secretary molecules to support “Well-known MAMPs are bacterial polysaccharides, surface proteins, PGNs, and secretory molecules [56].”

13) MAMPs, lines 169-170, p5: Specify receptors, and add references to support “These MAMPs can be sensed by various host PRRs and G-protein coupled receptors (GPCRs).”

14) MAMPs, lines 171-175, p5: Specify location, function, ligand specificity, and references (review coverage was insufficient) to support “Indeed, there are many host PRRs classified according to their location, function, and ligand specificity [57]. Based on their displayed pattern, each host receptor responds to its specific bacterial ligand, subsequently producing anti- or pro-inflammatory cytokines and chemokines to counteract against invadingmicrobes [58].”

15) MAMPs, lines 175-176, p5: Specify bacteria in “It has been reported that bacteria and their MAMPs could also affect osteoimmunology extensively [59].”

16) LPS, title, line 179, p5: Replace LPS by its full name in the title.

17) LPS, lines 180-182, p5: IJMS is focused on molecular science. Add chemical structure to support “LPS, also known as endotoxin, is a characteristic cell wall component of Gram-negative bacteria. It is composed of a hydrophobic lipid A, a hydrophilic core polysaccharide, and a hydrophilic O antigen-specific side polysaccharide chain [60].”

18) LPS, lines 182-183, p5: Add references to support “Lipid A, an anchoring part of LPS on bacterial outer membrane, plays a crucial role in inducing host immune responses.”

19) LPS, lines 184-185, p5: Add references, specify structure of antigen to support “O antigen is a sequential sugar molecule, which varies from bacteria to bacteria.”

20) LPS, lines 195-196, p5: Add references to support “To date, many reports have demonstrated that LPS induces bone loss at various sites in vivo.”

21) LPS, lines 200-201, p5: Add references to support “Overall numerous research indicate that LPS in- 200 duces bone loss under various physiological conditions.”

22) LPS, lines 203-205, p5: Add references, and specify cell origin, type, to support “LPS inhibits osteoclast differentiation when they are treated on the pre-osteoclast stage, but it triggers osteoclastogenesis when treated during committed osteoclast stage.”

23) LPS, lines 205-207, p5: Add references, specify cell origin, type, to support “Pre-osteoclasts treated with M-CSF and RANKL form osteoclasts, while pre-osteoclasts treated with M-CSF and LPS 206 do not form osteoclasts, indicating that LPS cannot be substituted for RANKL.”

24) LPS, lines 207-208, p5: Specify cell origin, type, add explanations, and rephrase “Furthermore, pre-osteoclasts treated with M-CSF, RANKL, and LPS show abolished osteoclastogenesis [74].”

25) LPS, from line 209, p5 to line 210, p6: Specify cell origin, type, add references to support “In contrast, committed osteoclasts show different patterns from pre-osteoclasts.”

26) LPS, lines 212-214, p6: Specify cell origin, type, add explanations, and references to support “In consideration of bone loss effects by LPS in vivo, it is likely that committed osteoclasts, rather than pre-osteoclasts, are more suitable to represent in vivo situation.”

27) LPS, lines 215-216, p6: Specify cell origin, type, and add references to support “Meanwhile, diminished osteoclastogenesis by TLR ligands seems to be a common phenomenon that occurs in pre-osteoclasts.”

28) LPS, lines 216-218, p6: Specify cell origin, type, and add references to support “When TLR ligands are treated, pre-osteoclasts (macrophages) perceive this as an infection situation and may preferentially cause macrophages to execute the host defense strategy rather than inducing osteoclast differentiation.”

29)LTA, Title, line 228, p6: Replace LTA by its full name in the title.

30) LTA,lines 229-231, p6: Add chemical structures to support “Based on its chemical structure, LTA is classified into five types (type I to V) and each bacterium has a distinct, characteristic LTA structure [84,85].”

31) LTA,lines 234-237, p6: Add molecular mechansims to support “LTA specifically attaches to the host cells through TLR2 and CD14, leading to MAPK and NF-B activation [86]. Consequently, the downstream cascade induces innate immune responses, such as production of nitric oxide and TNF-α [87,88].”

32) LTA,line 238, p6: Specify cell origin, type, and add references to support “In addition, LTA attenuates osteoclast differentiation.”

33) LTA, lines 247-248, p6: Specify animal mode, and add explanations since It was unclear how LTA from S. aureus promotes the synthesis of bone bridge, ossification, and healing of femoral fractures in “Additionally, LTA from S. aureus promotes the synthesis of bone bridge, ossification, and healing of femoral fractures [93].”

34) LPP, title line 252, p6: Replace LPP by its full name in the title.

35) LPP, line 255, p6: Specify lipid moiety in “The structure of LPP consists of a protein with a lipid moiety [95].”

36) LPP, from line 259, p6 to line 262, p7: Add a scheme to illustrate “In addition, TLR2 forms a heterodimer with TLR1 or TLR6, recognizing LPP. Diacylated LPPs are sensed by TLR2/TLR6 heterodimer, while triacylated LPPs are recognized by TLR1/TLR2 [98,99].”

37) LPP, lines 273-275, p7: Add explanations Pam2CSK4- or Pam3CSK4-soaked collagen sponge in “The calvaria from mouse was destructed by the inser-tion of Pam2CSK4- or Pam3CSK4-soaked collagen sponge [101].”

38) LPP, lines 279-280, p7: Specify cell origin, type, and add references and explanations to support “Consequently, aforementioned reports indicate that bacterial LPPs induce osteoclast formation and bone resorption via TLR2/MyD88 pathway.”

39) LPP, lines 284-285, p7: Add references to support “There are a number of carbohydrate and 284 protein adhesins in both Gram-positive and Gram-negative bacteria.”

40) LPP, lines 294-295, p7: Specify cell origin or animal model, and add references to support “Overall, bacterial fimbriae can induce bone resorption predominantly by inducing osteoclastogenesis.”

41) PGN, line 298, title, p7: Replace PGN by its full name in the title.

42) PGN, lines 299-303, p7: Add molecular structures to support “It is made up of polymers composed of N-acetylglucosamines (NAGs) and N-acetylmuramic acids (NAMs). Each NAM has a short peptide chain that is involved in forming a cross-linked peptide bridge between polymersMuramyl dipeptide (MDP; NAM-L-Ala-D-Glu) is a minimal essential structural motif of PGNs in both Gram-positive and Gram-negative bacteria [112]. ”

43) PGN, lines 305-306, p7: Add references to support “There are nucleotide-binding oligomerization domain (NOD) in the host cytoplasm, which is responsible for sensing PGN motifs.”

44) PGN, from line 308, p7 to 314, p8: Add scheme to support “Among the NODs, 308 NOD1 recognizes meso-DAP of Gram-negative bacterial PGNs, whereas NOD2 senses 309 MDP moieties of ubiquitous bacterial PGNs [114]. Once meso-DAP and MDP are recognized by NOD1 and NOD2, respectively, both NODs induce CARD-CARD interaction and then form the complex with the adaptor molecules, receptor-interacting protein-like interacting caspase-like apoptosis regulatory protein kinase (RICK), leading to NF-kappaB and MAPK activation for triggering inflammatory responses [114-116].”

45) PGN, line 346, p8: Table 1 was never cited and commented in the whole text.

46) SCFA, title, line 348, p9: Replace SCFA by its full name in the title.

47) SCFA, lines 354-357, p9: Indicate structural information on binding site of distinct types GPCRs, and specify their differences to support “Among GPCRs, SCFAs can bind and activate GPCR 40, 41, and 43, which are designated as free fatty acid receptor (FFAR) 1, 3, and 2, respectively, or GPCR 84, 120 and 109a [130,131].”

48) SCFA, lines 367-368, p9: Add references to support “SCFAs also affect bone metabolism by the regulation of osteoclasts and osteoblasts via GPCR or HDAC inhibition”

49) SCFA, lines 369-372, p9: Explain the relation between TRAP and FFAR1 in “Iwami et al. reported that sodium butyrate potently attenuates the formation of TRAP-positive multinucleated cells, which are cultured from bone marrow cells [139]. Further studies revealed that FFAR1 knock-out mice show less bone density than wild-type mice.”

50) SCFA, lines 385-387, p9: Add references to support “Collective studies indicated that SCFAs inhibit the osteoclastogenesis from pre-osteoclasts via HDAC inhibition and/or FFAR1 or 2 activation but not from committed osteoclast.”

51) Rearrange the whole paragraph since it was mixed with cell line, diseases animal or human studies in “Indeed, low concentration of butyrate induces histone H3 acetylation with concurring expression of ALP, osteonectin, and OPG in MG-63 osteoblastic cell line [144]. In contrast, high concentration of sodium butyrate inhibits the osteoblast differentiation and mineralization via suppression of osteoblast-specific factors, such as Runx2, osterix, and Dlx5 [145]. In fact, treatment of sodium butyrate at 16 mM attenuates osteoblast proliferation by suppression of cell cycle [146]. Although SCFAs show controversial effects on osteoblasts, they have the potential to treat destructive bone diseases. For instance, SCFAs or high fiber diet causing high SCFAs augment systemic bone density [147]. Moreover, SCFAs prevent bone loss against postmenopausal or inflammatory conditions [148].”

52) EV, Title, line 400, p10: Replace EV by its full name in the title.

53) EV, lines 401-404, p10: Add more information on different types of EVs in “EVs could be released from archaea, eukaryote, and bacteria [149]. Among them, the diameter of bacterial EVs is roughly 20 ~ 500 nanometers, and these spherical membrane- enveloped particles are secreted from parental bacteria into the extracellular environment [150].”

54) EV, lines 410-414 , 10: It was unclear if it were in vivo or in vitro experiments to support “In addition, F. alocis 410 EVs potently down-regulate osteogenic factors, such as Runx2, osterix, ALP, OCN, and 411 type I collagen, thereby attenuating bone mineralization. Notably, F. alocis EVs activate 412 TLR2 but not TLR4, and the inhibitory effect of F. alocis EVs on osteogenic differentiation 413 is fully dependent on TLR2 signaling pathway which mediate the activation of MAPK and 414 NF-κB [154].”

55) EV, lines 416-417, 10: It Add references to support “However, emerging evidence indicates that bacterial EVs might directly or indirectly influence osteoclast differentiation.”

56) EV, lines 417-419, 10: Specify types and origin of immunological cells in “Bacterial EVs ac- 417 tivate the immune cells to induce pro-inflammatory cytokines, such as IL-1β, IL-6, or TNF- 418 α, which trigger the osteoclast differentiation from committed osteoclasts [155,156].”

57) Extracellular polysaccharides, lines 430-433, p10: Add references, specify types, and origin of immunological cells in “For example, EPS purified from Bifidobacterium 431 longum (EPS-624) inhibit osteoclast differentiation from pre-osteoclasts by activating 432 TLR2 signaling pathway.”

58) Extracellular polysaccharides, lines 433-434, p10: Add references, specify types, and origin of osteoblasts in “In addition, EPS-624 show increased differentiation patterns in osteoblasts.”

59) Extracellular polysaccharides, lines 435-437, p10: Add references, specify if it was performed in vivo in “Moreover, EPS isolated from Vibrio diabolicus, which are hyaluronic acid-like EPS, potently enhance bone healing without abnormal bone growth.”

60) Extracellular polysaccharides, lines 438-440, p10: Add references, specify if it was performed in vivo in “In contrast, oversulfated EPS produced by Altermonas infernus (OS-EPS) inhibit the proliferation and mineralization activity of osteoblasts.”

61) Extracellular polysaccharides, lines 438-440, p10: Add references, specify origin of cells to support “Also, OS-EPS decrease the RANKL-induced osteoclast differentiation from pre-osteoclasts while increasing the collagenolytic activity of osteoclasts by using cathepsin K.”

62) Extracellular polysaccharides, lines 444-450, p10: Rephrase since it was unclear if there were mixed in vitro and in vivo experiments in “Similar to EPS, CPS also affect the activation and differentiation of bone cells, including osteoclasts and osteoblasts. CPS from Aggregatibacter actinomycetemcomitans Y4 (Aa-CPS) enhance the formation of osteoclasts and promote the bone resorptive activity through the induction of IL-1α [164]. Aa-CPS also upregulate PGE2, which has a positive effect on osteoclast formation [165]. These reports indicate that Aa-CPS-induced PGE2 and 4IL-1α are involved in inflammatory bone diseases such as periodontitis by promoting osteoclastogenesis.

63) Extracellular polysaccharides, from line 450, p10 to line 453, p11: The paragraph was mixed with in vitro and in vivo experiments and shall be rearranged “In addition, Aa-CPS show anti-proliferative activity by causing Fas-me- Moreover, immunization of CPS from P. gingivalis exhibits immunoglobulin responses and protects P. gingivalis-induced oral bone loss, suggesting that CPS are one of the responsible molecules for bone diseases [167]”

64) CDN, line 466, p11:Replace CDN by its full name in the title.

65) CDN, lines 467-469, p11:Add structural scheme to support “Stimulator of IFN genes (STING), also known as transmembrane 173, has 4 trans- 467 membrane regions and is located at the endoplasmic reticulum of host cell [173]. STING 468 directly recognizes the cytosolic CDNs, leading to secretion of type I IFNs [174].”

66) CDN, lines 478-486, p11: Specify cell types, and origin in “CDNs inhibit RANKL-induced osteoclast differentiation from pre-osteoclasts through STING-dependent signaling pathway [179]. They demonstrated that CDNs potently trigger STING-TBK1-IRF3 cascade and induce the mRNA expression of IFN-β during osteoclast differentiation. IFN-β in turn acts as a negative regulator of osteoclast differentiation by activating Jak-STAT signaling [180], which is the major pathway responsible for the inhibition of osteoclastogenesis. In contrast, because ubiquitin-mediated degradation of Jak in committed osteoclasts, CDNs- induced IFN-β cannot activate the Jak-STAT signaling pathway during differentiation of committed osteoclasts. Notably, CDNs prevent the RANKL-induced bone destruction in vivo [179].”

67) CDN, lines 487-488, p11: Specify cell types, and origin in “Recently, it has been reported that STING interacts with TRAF6 which is a TLR 487 signaling mediator [181].”

68) Table 2, p11: Table 2 was never citer and commented in the whole text.

69) Therapeutics, lines 494-495 p12: Add references to support “On the basis of information above, microbes influence bone metabolism by constant interaction with host using their various MAMPs.”

70) Therapeutics, lines 498-502, p12: Add references to support “Nevertheless, the emergence of antibiotic-resistant bacteria and remaining MAMPs after treatment pose significant challenge for complete clearance. Therefore, further studies are needed to understand the role of MAMPs in bone diseases and to control the immune responses induced by MAMPs.”

71) Therapeutics, lines 503-506, p12: Rephrase, since the whole paragraph is not directly related to bacteria-causing diseases in “Many therapeutic drugs, such as bisphosphonates, monoclonal antibodies, or hormone preparations, are developed to treat bone diseases by inhibiting bone resorption or inducing bone formation [183-185]. However, conventional drugs show unexpected side effects, such as nausea, osteonecrosis of jaw, or osteosarcoma [185-187].”

72) Therapeutics, lines 506-509, p12: It was unclear if the findings resulted from in vivo or in vitro experiments to support “Notably, several studies investigated that some MAMPs, especially derived from probiotics, decrease bone resorption or enhance bone formation by controlling the differentiation of osteoclasts or osteoblasts, respectively [122,148,179].”

73) Treatment of MAMPs-induced bone diseases, line 513, Title, p12: Replace MAMPs by its full name in the title.

74) Treatment of MAMPs-induced bone diseases, line 514-517, p12: Specify cell model and their origins to support “Most MAMPs are potent inducers of pro-inflammatory cytokines, such as IL-1, IL-6, 514 or TNF-α, via the recognition by PRRs [188]. These MAMP-induced pro-inflammatory cytokines positively influence the differentiation of committed osteoclasts into mature osteoclasts and the activity of osteoclasts, leading to bone loss [189].”

75) Treatment of MAMPs-induced bone diseases, line 519-523, and lines 524-527, p12: It was unclear which cell or animal models were used to support “For example, through the use of blocking antibodies that antagonize TNF and IL-1 receptors, significant reduction of inflammation and bone loss was observed concordant with osteoclastogenesis inhibition [101]. Antibody specific to IL-6 or IL-6 receptor directly inhibits osteoclast differentiation and restores bone loss [101,190].” And in “Since MAMPs directly promote osteoclast differentiation through the activation of PRR signaling pathway, its regulation using inhibitors or antibodies may also be effective to control MAMP-induced osteoclastogenesis [101,192].”

76) Treatment of MAMPs-induced bone diseases, line 528-531, p12: Add references to support “In the case of osteoblasts, targeting and inhibiting some potent MAMPs involved in diminishing osteoblasts could be a useful way to alleviate bone diseases. For instance, hindering the action of LPS, which is a potent osteoblast inhibitor, might be valuable to prevent LPS-induced bone loss at bacterial infections. There are several ways to inhibit the action of LPS.”

77) Treatment of MAMPs-induced bone diseases, line 532-333, and lines 538-539, p13: It was unclear if it was in vivo or in vitro finding to support “Jung et al. demonstrated that TLR4 decoy receptor inhibits LPS-induced NF-κB activation and prevents Gram-negative bacterial sepsis [193].” And in “Antitoxin peptide Pep 19-2.5, which is designed to bind to LPS, reduced TNF-α expression and inflammation [195]. In addition, Polymyxin B, which neutralizes LPS, shut down NF-B signaling pathway [196].”

78) Probiotics as therapeutic agent for bone health, lines 550-552 p13, lines 558-560, p13: Indicate if studies were from human or animal origin to support “For instance, supplementation of probiotics, microorganisms that offer health benefit to the host, prevent bone loss [198,199].” And in “Furthermore, L. rhamnosus GG, L. gasseri SBT2055, or L. brevis CD2 downregulates alveolar bone loss and inflammation in periodontitis model [205-207].”

79) Probiotics as therapeutic agent for bone health, line 566, p13: Table 3 was ner cited and commented in the whole text.

80) Probiotics as therapeutic agent for bone health, lines 570-572, p14: It was unclear if both reports used the same osteoporosis mouce model to support “Other NOD2 ligands, including M-TriLYS, L18-MDP, and murabutide, also promote osteoblast differentiation and diminish bone resorption [121,122].”

81) Probiotics as therapeutic agent for bone health, lines 580-581, p14: Specify cellular or animal model to support “SCFAs directly decrease the formation and function of osteoclasts by HDAC 580 inhibition and change the cellular metabolic condition [148].”

82) Probiotics as therapeutic agent for bone health, lines 581-582, p14: Specify cellular or animal model to support “In addition, adequate dose 581 of SCFAs upregulates ALP activity of osteoblasts [139]. Low dose of SCFAs increases osteoblast differentiation by HDAC inhibition [144]. “

83) Probiotics as therapeutic agent for bone health, lines 582-585, p14: Specify animal model to support “Moreover, SCFAs indirectly increase osteoblast differentiation in which butyrate-activated T cells release the osteoblast differentiation factor, Wnt10b [147,209]. Furthermore, SCFAs increase bone density systemically in postmenopausal or inflammatory bone loss model [148].”

84) Probiotics as therapeutic agent for bone health, lines 585-588, p14: Specify cellular or animal model to support Like SCFAs, CDNs, secretory bacterial second messengers, also decrease osteoclast differentiation via STING-mediated IFN-β signaling pathway. In addition, CDNs potently prevent RANKL-bone loss [179].

85) Conclusion, lines 594-595, p14: Delete “Bone homeostasis is maintained by the balance between osteoclast and osteoblast activity.”

86) Conclusion, lines 595-600, p14: The conclusion is too broad, rephrase “Recent studies have shown that bacteria are one of the unignorable factors for bone Bacterial MAMPs including cell wall components such as polysaccharide, surface proteins, and PGN, and secretory microbial molecules including SCFAs, EVs, EPS, and CDNs directly or indirectly modulate osteoclast and osteoblast activity. Therefore, bacterial MAMPs and secretory microbial molecules can be used as molecular targets for  bone-related diseases such as osteoporosis and periodontal diseases.”

Round 2

Reviewer 2 Report

General comments:

1) There are still several concerns which were not adequately addressed. There are mostly centered on the origin of cells, types of animals, animal models, which were often insufficiently delineated. The reason why I insist on this point is that, completely distinct effects are observed depending on the cell types or animal origins. (see minor comments, N° 22, 23, 24, 25, 26, 28, 38, 58, 66, 74, 75, 77, 81,and 84)

2) The overall manuscript needs to be much more clearly separated in cell models, animal and clinical findings.

Minor comments:

22) LPS, lines 203-205, p5: Add references, and specify cell origin, type, to support “LPS inhibits osteoclast differentiation when they are treated on the pre-osteoclast stage, but it triggers osteoclastogenesis when treated during committed osteoclast stage.”

Author’s answer : As suggested, we added a reference in the revised manuscript.

Reviewer’s answer: It was partially answered, cell origin (which animal ?) was unspecified.

23) LPS, lines 205-207, p5: Add references, specify cell origin, type, to support “Pre- osteoclasts treated with M-CSF and RANKL form osteoclasts, while pre-osteoclasts treated with M-CSF and LPS 206 do not form osteoclasts, indicating that LPS cannot be substituted for RANKL.”

Author’s answer: As suggested, we added a reference and specified cell origin in the revised manuscript. (Line 246 at Page 7)

Reviewer’s answer: It was partially answered, cell origin (which animal ?) was unspecified.

24) LPS, lines 207-208, p5: Specify cell origin, type, add explanations, and rephrase “Furthermore, pre-osteoclasts treated with M-CSF, RANKL, and LPS show abolished osteoclastogenesis [74].”

Author’s answer: As suggested, we replaced the sentence into another one. (Line 252 at Page 7) “

Reviewer’s answer: It was partially answered, cell origin (which animal ?) was unspecified.

25) LPS, from line 209, p5 to line 210, p6: Specify cell origin, type, add references to support “In contrast, committed osteoclasts show different patterns from pre-osteoclasts.”

Author’s answer: As suggested, we added a reference.

Reviewer’s answer: It was partially answered, cell origin (which animal ?) was unspecified.

26) LPS, lines 212-214, p6: Specify cell origin, type, add explanations, and references to support “In consideration of bone loss effects by LPS in vivo, it is likely that committed osteoclasts, rather than pre-osteoclasts, are more suitable to represent in vivo situation.”

Author’s answer: As suggested, we added a reference for in vivo situation.

Reviewer’s answer: It was partially answered, cell origin (which animal ?) was unspecified.

28) LPS, lines 216-218, p6: Specify cell origin, type, and add references to support “When TLR ligands are treated, pre-osteoclasts (macrophages) perceive this as an infection situation and may preferentially cause macrophages to execute the host defense strategy rather than inducing osteoclast differentiation.”

Author’s answer: As suggested, we added a reference and cell origin of pre-osteoclasts. (Line 263 at Page 7) “

Reviewer’s answer: Reviewer’s answer: It was partially answered, cell origin (which animal ?) was unspecified.

38) LPP, lines 279-280, p7: Specify cell origin, type, and add references and explanations to support “Consequently, aforementioned reports indicate that bacterial LPPs induce osteoclast formation and bone resorption via TLR2/MyD88 pathway.”

Author’s answer: As suggested, we added some explanations and references.

Reviewer’s answer: Reviewer’s answer: It was partially answered, cell origin (which animal ?) was unspecified.

58) Extracellular polysaccharides, lines 433-434, p10: Add references, specify types, and origin of osteoblasts in “In addition, EPS-624 show increased differentiation patterns in osteoblasts.”

Author’s answer: As suggested, we inserted the specific information about cells and the reference in the revised manuscript.

Reviewer’s answer: It was partially answered, cell origin (which animal ?) was unspecified.

66) CDN, lines 478-486, p11: Specify cell types, and origin in “CDNs inhibit RANKL- induced osteoclast differentiation from pre-osteoclasts through STING-dependent signaling pathway [179]. They demonstrated that CDNs potently trigger STING-TBK1-IRF3 cascade and induce the mRNA expression of IFN-β during osteoclast differentiation. IFN-β in turn acts as a negative regulator of osteoclast differentiation by activating Jak-STAT signaling [180], which is the major pathway responsible for the inhibition of osteoclastogenesis. In contrast, because ubiquitin-mediated degradation of Jak in committed osteoclasts, CDNs- induced IFN-β cannot activate the Jak-STAT signaling pathway during differentiation of committed osteoclasts. Notably, CDNs prevent the RANKL-induced bone destruction in vivo [179].”

Author’s answer: As suggested, we inserted the specific information about origin of cells and the reference in the revised manuscript.

Reviewer’s answer: It was partially answered, cell origin (which animal ?) was unspecified.

74) Treatment of MAMPs-induced bone diseases, line 514-517, p12: Specify cell model and their origins to support “Most MAMPs are potent inducers of pro-inflammatory cytokines, such as IL-1, IL-6, or TNF-α, via the recognition by PRRs [188]. These MAMP-induced pro- inflammatory cytokines positively influence the differentiation of committed osteoclasts into mature osteoclasts and the activity of osteoclasts, leading to bone loss [189].”

Author’s answer: As suggested, we added the information of cells in the revised manuscript.

Reviewer’s answer : It was partially answered, cell origin (which animal ?) was unspecified.

75) Treatment of MAMPs-induced bone diseases, line 519-523, and lines 524-527, p12: It was unclear which cell or animal models were used to support “For example, through the use of blocking antibodies that antagonize TNF and IL-1 receptors, significant reduction of inflammation and bone loss was observed concordant with osteoclastogenesis inhibition [101]. Antibody specific to IL-6 or IL-6 receptor directly inhibits osteoclast differentiation and restores bone loss [101,190].” And in “Since MAMPs directly promote osteoclast differentiation through the activation of PRR signaling pathway, its regulation using inhibitors or antibodies may also be effective to control MAMP-induced osteoclastogenesis [101,192].”

Author’s answer: We inserted the information as suggested in the revised manuscript.

Reviewer’s answer: It was partially answered, it was unclear which types of cells, from which origins served to obtain in vitro experiments and in which animal for the in vivo experiments.

77) Treatment of MAMPs-induced bone diseases, line 532-333, and lines 538-539, p13: It was unclear if it was in vivo or in vitro finding to support “Jung et al. demonstrated that TLR4 decoy receptor inhibits LPS-induced NF-κB activation and prevents Gram-negative bacterial sepsis [193].” And in “Antitoxin peptide Pep 19-2.5, which is designed to bind to LPS, reduced TNF-α expression and inflammation [195]. In addition, Polymyxin B, which neutralizes LPS, shut down NF-κB signaling pathway [196].”

Author’s answer: We added the information as suggested in the revised manuscript.

Reviewer’s answer: It was partially answered, it was unclear which types of cells, from which origins served to obtain in vitro experiments and in which animal for the in vivo experiments.

81) Probiotics as therapeutic agent for bone health, lines 580-581, p14: Specify cellular or animal model to support “SCFAs directly decrease the formation and function of osteoclasts by HDAC inhibition and change the cellular metabolic condition [148].”

Author’s answer: As suggested, we inserted the information in the revised manuscript.
(Line 713 at Page 20) “SCFAs directly decrease the formation and function of osteoclasts by HDAC inhibition and change the cellular metabolic condition in vitro [114].”

Reviewer’s answer: It was partially answered, it was unclear which types of cells, from which origins served to obtain in vitro experiments.

84) Probiotics as therapeutic agent for bone health, lines 585-588, p14: Specify cellular or animal model to support “Like SCFAs, CDNs, secretory bacterial second messengers, also decrease osteoclast differentiation via STING-mediated IFN-β signaling pathway. In addition, CDNs potently prevent RANKL-bone loss [179].”

Author’s answer: As suggested, we added the information in the revised manuscript.

(Line 722 at Page 20) “Like SCFAs, CDNs which are secretory bacterial second messengers also decrease osteoclast differentiation  in vitro  via STING-mediated IFN-β signaling pathway.”

Reviewer’s answer: It was partially answered, it was unclear which types of cells, from which origins served to obtain in vitro experiments.

Round 3

Reviewer 2 Report

The authors addressed adequately the concerns

Author Response

Reviewer 2

Comments and Suggestions for Authors

The authors addressed adequately the concerns

 We appreciate the Reviewer’s compliment.